# Nested Dithered Quantization for Communication Reduction in Distributed Training

## Abstract

In distributed training, the communication cost due to the transmission of gradients or the parameters of the deep model is a major bottleneck in scaling up the number of processing nodes. To address this issue, we propose *dithered quantization* for the transmission of the stochastic gradients and show that training with *Dithered Quantized Stochastic Gradients (DQSG)* is similar to the training with unquantized SGs perturbed by an independent bounded uniform noise, in contrast to the other quantization methods where the perturbation depends on the gradients and hence, complicating the convergence analysis. We study the convergence of training algorithms using DQSG and the trade off between the number of quantization levels and the training time. Next, we observe that there is a correlation among the SGs computed by workers that can be utilized to further reduce the communication overhead without any performance loss. Hence, we develop a simple yet effective quantization scheme, nested dithered quantized SG (NDQSG), that can reduce the communication significantly *without requiring the workers communicating extra information to each other*. We prove that although NDQSG requires significantly less bits, it can achieve the same quantization variance bound as DQSG. Our simulation results confirm the effectiveness of training using DQSG and NDQSG in reducing the communication bits or the convergence time compared to the existing methods without sacrificing the accuracy of the trained model.

## 1 Introduction

In recent years, the size of deep learning problems has increased significantly both in terms of the number of available training samples as well as the complexity of the model. Hence, training deep models on a single processing node is unappealing or nearly impossible. As such, large-scale distributed machine learning in which the training samples are distributed among different repository or processing units (referred to as workers) has started to be a viable approach for tackling the memory, storage and computational constraints.

The requirement to exchange the gradients or the parameters of the model incurs significant communication overhead which is a major bottleneck in distributed training algorithms. In recent years, there has been a great amount of effort on reducing the communication overhead. The majority of existing methods can be categorized into two groups: The first group mitigates the communication bottleneck by reducing the overall transmission rate via sparsification, quantization and/or compression of the gradients. For example, Seide et al. (2014) reduces the communication overhead significantly by one-bit quantization of the stochastic gradients (SG). However, the reduced accuracy of gradient may impair the convergence rate. Using different quantization levels or adaptive quantizers, one can alleviate such issues by decreasing the error in the quantized gradients in the expense of increased communication bits Dryden et al. (2016). Moreover, applying entropy coding algorithms such as Huffman coding on the quantized values can further reduce the communication bit-rate Øland & Raj (2015); Strom (2015). Alistarh et al. (2017) introduced QSGD which uses probabilistic (stochastic) quantization of SGs instead of ordinary fixed (deterministic) quantization methods. They investigated its convergence guarantee and the trade-off between the quantization precision and variance of QSG. Terngrad Wen et al. (2017) probabilistically quantizes the gradients into $\{-1, 0, +1\}$ and it is shown that the convergence rate can be improved by layer-wise quantization and gradient clipping.

The second group of works attempts to attenuate the communication bottleneck by relaxing the synchronization between workers. Each worker may continue its own computations while some others are still communicating and exchanging parameters. Carefully scheduling and managing the asynchronous parameter exchange can lead to a better utilization of both the communication bandwidth and the computational power of the distributed system. Examples of such approaches include DownpourSGD Dean et al. (2012), Hogwild! Niu et al. (2011), Hogwild++ Zhang et al. (2016) and Stale Synchronous Parallel model of computation Ho et al. (2013).

**Our Contributions.** Our work in this paper falls within the first line of research, i.e. reducing the communication overhead by quantizing and compressing the gradients. We first introduce using *dithered quantization* in the distributed computations of the stochastic gradient and show that stochastic quantizer of Alistarh et al. (2017) and ternarization of Wen et al. (2017) can be considered as special cases of our proposed method, although the reconstruction algorithms are slightly different. The convergence of dithered quantized stochastic gradient descent algorithm is analyzed and its convergence speed w.r.t. the number of workers and quantization precision is investigated. Next, we observe that in a typical distributed system, the stochastic gradients computed by the workers are correlated. However, the existing communication methods ignore that correlation. We tap into the question of how that correlation can be exploited to further reduce the communication without sacrificing the precision or convergence of the learning algorithm. We model the correlation between the stochastic gradients computed by each worker and propose a *nested quantization* scheme to reduce the communication bits without increasing the variance of the quantization error or reducing the convergence speed of the distributed training algorithm.

## 1.1 NOTATIONS

Throughout the paper, bold lowercase letters represent vectors and the $i$-th element of the vector $\boldsymbol{x}$ is denoted as $x_i$. Matrices are denoted by bold capital letters such as $\boldsymbol{X}$, with the $(i, j)$-th element represented by $X_{i,j}$ or $[\boldsymbol{X}]_{i,j}$. Given a real number $x \in \mathbb{R}$, $\lfloor x \rceil$ is the nearest integer to $x$. For a random variable $u$, $u \sim \mathcal{U}[a, b]$ if its probability distribution is uniform over interval $[a, b]$ and $u \sim \mathcal{N}(\mu, \sigma^2)$ if it follows a Gaussian distribution with mean $\mu$ and variance $\sigma^2$.

## 2 PRELIMINARIES

### 2.1 DITHERED QUANTIZATION

It is well-known that the error in ordinary quantization especially when the number of quantization levels is low, depends on the input signal and is not necessarily uniformly distributed. In *Dithered Quantization*, a (pseudo-)random signal called dither is added to the input signal prior to quantization. Adding this controlled perturbation can cause the statistical behavior of the quantization error to be more desirable Schuchman (1964); Gray & Stockham (1993); Gray & Neuhoff (1998).

Let $Q(\cdot)$ be an M-level uniform quantizer with quantization step size of $\Delta$, i.e., $Q(v) = \Delta \lfloor v/\Delta \rceil$ where $\lfloor \alpha \rceil$ is the nearest integer to $\alpha$. The dithered quantizer is defined as follows;[1]

**Definition** (Dithered Quantization). For an input signal $x$, let $u$ be a dither signal, independent of $x$. The dithered quantization of $x$ is defined as $\tilde{x} = Q(x + u) - u$.

*Remark* 1. To transmit the dithered quantization of $x$, it is sufficient to send the index of the quantization bin that $x + u$ resides in, i.e., $\lfloor (x + u)/\Delta \rceil$. The receiver reproduces the (pseudo-)random sequence $u$ using the same random number generator algorithm and seed number as the sender. It is then subtracted from $Q(x + u)$ to form the dithered quantized value, $\tilde{x}$.

**Theorem 1** (Schuchman (1964)). *If 1) the quantizer does not overload, i.e., $|x + u| \leq \frac{M\Delta}{2}$ for all input signals $x$ and dither $u$, and 2) The characteristic function of the dither signal, defined as $M_u(j\nu) = \mathbb{E}_u[e^{j\nu u}]$, satisfies $M_u(j\frac{2\pi l}{\Delta}) = 0$ for all $l \neq 0$, then the quantization error $e = x - \tilde{x}$ is uniform over $[-\Delta/2, \Delta/2]$ and it is independent of the signal $x$.*

It is common to consider $\mathcal{U}[\Delta/2, \Delta/2]$ as the distribution of the random dither signal. It can be easily verified that this choice of the dither signal satisfies the conditions of Thm. 1, and it does not increase

---

[1]Throughout the paper, we assume that all quantizers are centered around 0. This is the case also for ternary Wen et al. (2017) and stochastic quantizations Alistarh et al. (2017).

the bound of the quantization error, i.e, $|\tilde{x} - x| \leq \Delta/2$ which is the same as the traditional uniform quantization with the same step size.

In some cases, the receiver may not be able to reproduce the dither signal to subtract from $Q(x + u)$. Hence, quantization is simply defined as as $\tilde{x}_h = Q(x + u)$. We refer to this approach as the *half-dithered quantization* as the dither signal is applied only to the quantizer, not the reconstruction of $x$. In this case, the quantization error is not necessarily independent of the signal, however by an appropriate choice of the dither signal, the moments of the quantization error will be independent Gray & Stockham (1993). For example, if the dither signal $u$ is the sum of $k$ independent random variables, each having uniform distribution $\mathcal{U}[-\Delta/2, \Delta/2]$, then the $k$-th moment of the quantization error, $\epsilon = x - \tilde{x}_h$, would be independent of the signal, given by $\mathbb{E}\left[\epsilon^k | x\right] = \mathbb{E}\left[\epsilon^k\right] = (k+1)\frac{\Delta^2}{12}$.

### 2.1.1 RELATIONSHIP WITH TERNARY AND STOCHASTIC QUANTIZATIONS

Here, we examine the relation between the dithered quantization, Ternary quantization of Wen et al. (2017) and the stochastic quantization in Alistarh et al. (2017). Without loss of generality, assume that the vector $\boldsymbol{x}$ is normalized such that $|x_i| \leq 1$. Although the reconstruction of quantized values in our method is different from those in TernGrad and QSGD, we show that these quantizers can be considered as a special case of the half-dithered quantizer.

$M$-level Stochastic Quantization in Alistarh et al. (2017) is defined as

$$Q^{(s)}(x_i) = \begin{cases} \text{sign}(x_i)\, l/M & \text{with probability } l + 1 - M|x_i| \\ \text{sign}(x_i)\,(l+1)/M & \text{with probability } M|x_i| - l \end{cases}, \tag{1}$$

where $|x_i| \in [l/M, (l+1)/M]$. The ternary quantizer of Wen et al. (2017) can be considered as a special case of stochastic quantizer with $M = 1$.

**Lemma 2.** *Stochastic quantization is the same as $(2M + 1)$-level half-dithered quantizer with step-size $\Delta = \frac{1}{M}$ and uniform dither $u \sim \mathcal{U}[-\frac{1}{2M}, \frac{1}{2M}]$.*

In other words, stochastic quantizer adds a uniformly distributed dither to the input signal before quantization, but at the receiver, it *does not* subtract the dither from the quantized value. Therefore, the quantization error is not independent of the signal Gray & Stockham (1993). It can be easily verified that although the quantization is unbiased, $\mathbb{E}\left[\boldsymbol{x} - Q^{(s)}(\boldsymbol{x})\right] = \boldsymbol{0}$, its variance depends on the value of the input signal:

$$\mathbb{E}\left[([Q^{(s)}(\boldsymbol{x}) - \boldsymbol{x}]_i)^2\right] = (|x_i| - l/M)((l+1)/M - |x_i|), \quad \text{if } |x_i| \in [l/M, (l+1)/M].$$

It can be easily verified that the variance of the quantization error varies in the interval $[0, \frac{1}{4M^2}]$ depending on the value of $x$. If $x$ is uniformly distributed over $[-1, 1]$, the average quantization variance would be $\frac{1}{6M^2}$, twice the variance of the dithered quantization.

## 2.2 NESTED QUANTIZATION

Here, we briefly overview the definition and some properties of the nested quantization. Especially we focus on the one dimensional case as our algorithm is based on scalar quantization.

**Definition** (Nested Quantizers). The pair $(Q_1, Q_2)$ of two quantizers are nested if and only if $\forall \boldsymbol{x}$, $Q_1(Q_2(\boldsymbol{x})) = Q_2(\boldsymbol{x})$, but the opposite does not necessarily hold. $Q_1(\cdot)$ and $Q_2(\cdot)$ are called the fine and coarse quantizers, respectively.

As a result, the centers of the quantization bins in the coarse quantizer is a subset of those of the fine quantizer. In the one dimensional case, if $Q_1$ and $Q_2$ have quantization step sizes equal to $\Delta_1$ and $\Delta_2$, respectively, it can be easily verified that they are nested if and only if there exists a constant integer $k > 1$ such that $\Delta_2 = k\Delta_1$. For the definition and properties of higher di-

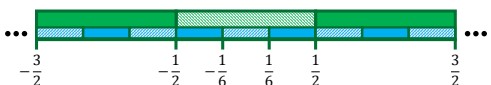

Figure 1: Nested one-dimensional quantizers, fine quantizer (blue) with $\Delta_1 = 1/3$ and coarse quantizer (green) with $\Delta_2 = 1$.

mensional nested quantization using lattices please refer to Zamir et al. (2002); Zamir (2009) and references therein.

## 3 Distributed Training Using Dithered Quantization

Let $\mathcal{W} \subset \mathbb{R}^n$ be a known set of possible parameters $\boldsymbol{w}$ and $\mathcal{L} : \mathcal{W} \to \mathbb{R}$ be a differentiable objective function to be minimized. A stochastic gradient $\boldsymbol{g}$ of $\mathcal{L}(\boldsymbol{w})$ is an unbiased random estimator of the gradient, i.e., $\boldsymbol{g}$ is a random function such that $\mathbb{E}[\boldsymbol{g}] = \nabla_{\boldsymbol{w}}\mathcal{L}$. Specifically, if $\mathcal{L}(\boldsymbol{w}) = \mathbb{E}_{\boldsymbol{x} \in \mathcal{X}}[f(\boldsymbol{x}; \boldsymbol{w})]$, where $\mathcal{X}$ is the set of training data samples and $f(\boldsymbol{x}; \boldsymbol{w})$ is a smooth differentiable parametric function, then given a mini-batch $\{\boldsymbol{x}_1, \ldots, \boldsymbol{x}_L\}$ of training samples, the stochastic gradient of $\mathcal{L}(\boldsymbol{w})$ can be computed as $\boldsymbol{g} = \frac{1}{L}\sum_l \nabla_{\boldsymbol{w}} f(\boldsymbol{x}_l; \boldsymbol{w})$.

We consider the distributed training scenario shown in Fig. 2. There are $P$ separate workers (processing nodes) which have their own copy of the model to be trained. At each iteration of the training, each worker computes a stochastic gradient of the parameters $\boldsymbol{g}_k$, or the update in the parameters $\boldsymbol{\delta W}_k$, based on its own available data. It is then transmitted to a server (in the centralized training) or communicated with other workers (in the decentralized topology) to compute the average. The average of all gradients or the updates ($\bar{\boldsymbol{g}}$ or $\boldsymbol{\delta \bar{W}}$) is then broadcasted back to all workers. In the following, we focus on the distributed training using stochastic gradients with a centralized ag-

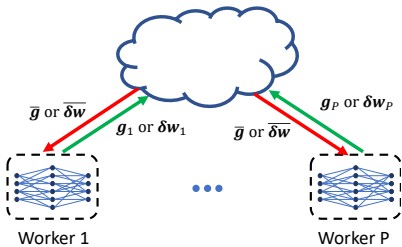

Figure 2: Schematic overview of the distributed training.

gregation node. First, we consider the use of dithered quantization in training and analyze the convergence of the learning algorithm in both single worker and distributed (multiple workers) training scenarios. Next, we observe that the stochastic gradients computed at the workers are correlated. We define a correlation model to capture the dependency between SGs of the workers and show that how nested dithered quantization can help further reducing the communication bits at each iteration of training without sacrificing the accuracy or the number of iterations to converge.

### 3.1 Dithered Quantized Stochastic Gradient

We consider the dithered quantization of SG (DQSG) as follows: Let $Q(\cdot)$ be a uniform quantizer with quantization step size $\Delta$, and $\boldsymbol{u} \sim \mathcal{U}[-\Delta/2, \Delta/2]$ be the random dither signal. The dithered quantized SG is given by

$$\tilde{\boldsymbol{g}} = \kappa\left(Q(\boldsymbol{g}/\kappa + \boldsymbol{u}) - \boldsymbol{u}\right), \tag{2}$$

where the *scale factor* $\kappa = \|\boldsymbol{g}\|_\infty = \max_i |g_i|$ maps the gradient into the range $[-1, 1]$. By Thm. 1, the scaled quantization noise $\boldsymbol{e} = (\boldsymbol{g} - \tilde{\boldsymbol{g}})/\kappa$ will be independent from $\boldsymbol{g}$ and uniformly distributed over $[-\Delta/2, \Delta/2]$. Note that by setting $\Delta = 1/M$, we will have a $2M + 1$ level quantizer with quantization bins' indexes in $\{-M, \ldots, -1, 0, 1, \ldots, M\}$.

**Lemma 3.** *Let $\boldsymbol{g}$ be a stochastic gradient of $\mathcal{L}(\boldsymbol{w})$. Then, the DQSG, $\tilde{\boldsymbol{g}}$, has the following properties:*

P1. *$\tilde{\boldsymbol{g}}$ is unbiased, i.e., $\mathbb{E}[\tilde{\boldsymbol{g}}] = \nabla_{\boldsymbol{w}}\mathcal{L}$,*

P2. *Its variance is bounded as $\mathbb{E}\left[\|\tilde{\boldsymbol{g}} - \nabla_{\boldsymbol{w}}\mathcal{L}\|_2^2\right] \leq \frac{n\Delta^2}{12}\mathbb{E}\left[\|\boldsymbol{g}\|_2^2\right] + \mathbb{E}\left[\|\boldsymbol{g} - \nabla_{\boldsymbol{w}}\mathcal{L}\|_2^2\right].$*

*Especially, if we assume that the difference between the stochastic gradients and the true ones behaves like a Gaussian noise, i.e., $\boldsymbol{g} - \nabla_{\boldsymbol{w}}\mathcal{L} = \boldsymbol{\nu}$ where $\boldsymbol{\nu} \sim \mathcal{N}(0, \sigma^2)^2$, then*

$$\mathbb{E}\left[\|\tilde{\boldsymbol{g}} - \nabla_{\boldsymbol{w}}\mathcal{L}\|_2^2\right] - \mathbb{E}\left[\|\boldsymbol{g} - \nabla_{\boldsymbol{w}}\mathcal{L}\|_2^2\right] \leq \frac{\Delta^2}{3}\ln(\sqrt{2}n)\,\mathbb{E}\left[\|\boldsymbol{g} - \nabla_{\boldsymbol{w}}\mathcal{L}\|_2^2\right] + \frac{n\Delta^2}{6}\|\nabla_{\boldsymbol{w}}\mathcal{L}\|_\infty^2. \tag{3}$$

As a result of Lemma 3, we observe that *the excess variance* caused by quantization is proportional to $\Delta^2$. Hence by adding 1 bit, i.e., doubling the number of quantization levels, it is reduced by a factor of 4. Further, we notice that how partitioning the stochastic gradient into $K$ sub-vectors can reduce the variance of DQSG at the expense of extra communication bits. Let $\tilde{\boldsymbol{g}}^K$ be the DQSG resulted from partitioning $\boldsymbol{g}$ into $K$ sub-vectors and quantizing them separately. For the simplicity of analysis assume that the partitions are of equal length, $n/K$. Simple calculations reveal that

$$\mathbb{E}\left[\|\tilde{\boldsymbol{g}}^K - \nabla_{\boldsymbol{w}}\mathcal{L}\|_2^2\right] - \mathbb{E}\left[\|\boldsymbol{g} - \nabla_{\boldsymbol{w}}\mathcal{L}\|_2^2\right] \leq \frac{\Delta^2}{6}\left[2\ln(\sqrt{2}\frac{n}{K})\mathbb{E}\left[\|\boldsymbol{g} - \nabla_{\boldsymbol{w}}\mathcal{L}\|_2^2\right] + n\|\nabla_{\boldsymbol{w}}\mathcal{L}\|_\infty^2\right] \tag{4}$$

---

[2]Usually, the SG is computed as $\boldsymbol{g} = \frac{1}{L}\sum_l \nabla_{\boldsymbol{w}} f(\boldsymbol{x}_l; \boldsymbol{w})$ and for large enough $L$, due to the central limit theorem, $\boldsymbol{g} - \nabla_{\boldsymbol{w}}\mathcal{L} \xrightarrow{d} \mathcal{N}(\boldsymbol{0}, \boldsymbol{\Sigma}/\sqrt{L})$ for an appropriate fixed covariance matrix $\boldsymbol{\Sigma}$.

The first term decreases logarithmically w.r.t. the number of partitions. On the other hand, each partition requires transmitting an additional scale factor ($\kappa$ in (2), see Alg. 1), incurring extra $Kb$ bits in total, where $b$ is the number of bits for each scale factor. Hence, the excess communication bits due to partitioning increases linearly, while the first term in the excess variance decreases logarithmically.

**Convergence Analysis.** We now analyze the convergence of the gradient descent algorithm with the dithered quantized stochastic gradients. At the $t$-th iteration, the parameters are updated as

$$\boldsymbol{w}_{t+1} = \boldsymbol{w}_t - \eta_t \widetilde{\boldsymbol{g}}_t, \tag{DQSGD}$$

where $\eta_t$ is the learning rate and $\widetilde{\boldsymbol{g}}_t$ is the DQSG.

Recall that $\widetilde{\boldsymbol{g}} = \boldsymbol{g} + \|\boldsymbol{g}\|_\infty \boldsymbol{\epsilon}$, where $\boldsymbol{\epsilon} \sim \mathcal{U}[-\Delta/2, \Delta/2]$ is the quantization noise, independent of $\boldsymbol{g}$. Hence, *training with dithered quantized SG is the same as training with non-quantized SG corrupted by an independent bounded uniform noise*. If the quantization step size and hence the noise is controlled appropriately, the quantization noise can improve the training of very deep models Neelakantan et al. (2015); Noh et al. (2017)

Moreover, analyzing the convergence of (DQSGD) is almost the same as the ordinary SGD. For example, since $\mathbb{E}\left[\|\widetilde{\boldsymbol{g}}\|_2^2\right] \leq \left(1 + n\frac{\Delta^2}{12}\right)\mathbb{E}\left[\|\boldsymbol{g}\|_2^2\right]$, under the same assumptions as of Bottou (1998), the convergence of DQSGD can be proven, which is replicated here for the sake of completeness.

**Theorem 4.** *Assume that i) $\mathcal{L}(\boldsymbol{w})$ has a single minimum, $\boldsymbol{w}^*$, ii) $\forall \epsilon > 0$, $\inf_{\|\boldsymbol{w}-\boldsymbol{w}^*\|_2 > \epsilon}(\boldsymbol{w} - \boldsymbol{w}^*)^\mathsf{T}\nabla_{\boldsymbol{w}}\mathcal{L} > 0$, iii) $\sum_t \eta_t = +\infty$ and $\sum_t \eta_t^2 < +\infty$, and iv) for some constants $A$ and $B$, stochastic gradients satisfy $\mathbb{E}\left[\|\boldsymbol{g}(\boldsymbol{w})\|_2^2\right] \leq A + B\|\boldsymbol{w} - \boldsymbol{w}^*\|_2^2$. Then for any quantization step size $\Delta \leq 1$, training with DQSGD converges to the solution almost surely.*

Next, we investigate how the number of workers and quantization step size affects the training time in the proposed distributed training scheme.

**Distributed Training with DQSGD.** Algorithm 1 summarizes the proposed distributed training with $P$ workers using dithered quantization of SG (DQSG). The $p$-th worker, first computes the stochastic gradient $\boldsymbol{g}_p$ and then using the scale parameter $\kappa_p = \|\boldsymbol{g}_p\|_\infty$, computes the *quantization index $\boldsymbol{q}_p$* (see Remark 1). Hence, the DQSG is given by $\widetilde{\boldsymbol{g}}_p = \kappa_p(\Delta.\boldsymbol{q}_p - \boldsymbol{u}_p)$. To be able to reproduce the (pseudo-)random sequences at the server, the same random number generator algorithm and seed number, $s_p$, is used at both the worker and the server. At each iteration of training, the seed numbers are updated according to a predetermined algorithm at all workers and the server, to prevent generating the same random sequences repeatedly.

Using the above distributed training algorithm, the following result on the convergence time of distributed (DQSGD) algorithm can be proved.

**Theorem 5.** *Let $\mathcal{W} \subset \mathbb{R}^n$ be a convex set and $\mathcal{L}(\boldsymbol{w})$ be a convex, Lipschitz-smooth function with constant $\ell$[3]. Further, assume that $\mathcal{L}$ achieves its minimum at $\boldsymbol{w}^*$ and has bounded gradients almost everywhere, i.e., for a constant $B > 0$, $\|\nabla\mathcal{L}\|_2 \leq B$.*

*Let the initial point for the learning algorithm be $\boldsymbol{w}_0$ and $R = \sup_{\boldsymbol{w} \in \mathcal{W}} \|\boldsymbol{w} - \boldsymbol{w}_0\|_2$. Consider distributed training algorithm (Alg. 1) on $P$ workers using (DQSGD) with quantization step size $\Delta$. Suppose that the workers can compute stochastic gradients with variance bound $V$, i.e,. $\mathbb{E}\left[\|\boldsymbol{g}_p - \nabla_{\boldsymbol{w}}\mathcal{L}\|_2^2\right] \leq V$. Define $\sigma^2 = V(1 + n\Delta^2/12) + nB\Delta^2/12$. Then for sufficiently small $\epsilon > 0$, after $T$ steps of training with constant step size $\eta_t$, where*

$$T = 2.5\frac{R^2}{\epsilon^2}\frac{\sigma^2}{P}, \quad \text{and} \quad \eta_t = \epsilon/(\epsilon\ell + 1.1\sigma^2/P),$$

*we have*

$$\mathbb{E}\left[\mathcal{L}\left(\frac{1}{T}\sum_{t=1}^T \boldsymbol{w}_t\right)\right] - \mathcal{L}(\boldsymbol{w}^*) \leq \epsilon.$$

Let $T_c$ be the training time without any quantization in the above setup. Then, it can be easily verified that the training time of the dithered quantization is increased by

$$\frac{T - T_c}{T_c} = \frac{n\Delta^2}{12}\left(1 + \frac{B}{V}\right). \tag{5}$$

---

[3]i.e., $\|\nabla\mathcal{L}(\boldsymbol{w}_1) - \nabla\mathcal{L}(\boldsymbol{w}_2)\|_2 \leq \ell\|\boldsymbol{w}_1 - \boldsymbol{w}_2\|_2$

---

**Algorithm 1:** Distributed Training Using Dithered Quantization of SG

---

**Initialization**
- Assign a random seed $s_p$ to the $p$-th worker and initialize the parameters with $\boldsymbol{w}_0$, $p = 1, 2 \ldots, P$.
- Keep a copy of $s_p$'s at the server.
- Set $\Delta$, the quantization step-size, and the associated uniform quantizer, $Q(\cdot)$.

**for** *each iteration of training* **do**
    **Workers** $p = 1, 2, \ldots, P$:
        - Get a batch of training data and compute the stochastic gradients $\boldsymbol{g}_p$.
        - Generate a pseudo-random sequence $\boldsymbol{u}_p$, uniformly distributed over $[-\Delta/2, \Delta/2]$ using seed $s_p$.
        - Compute the quantization index: $\boldsymbol{q}_p = \lfloor \boldsymbol{t}/\Delta \rfloor$ where $\boldsymbol{t} = \boldsymbol{g}_p/\kappa_p + \boldsymbol{u}_p$ and $\kappa_p = \|\boldsymbol{g}\|_\infty$.
        - Update the seed number $s_p$.
        - Send $\kappa_p$ and $\boldsymbol{q}_p$ (or the corresponding quantization bin).
    **Server** :
        - Reproduce the pseudo-random sequence $\boldsymbol{u}_p$ using the seed number $s_p$.
        - Reconstruct the gradient of the $p$-th worker as $\tilde{\boldsymbol{g}}_p = \kappa_p \left( \Delta.\boldsymbol{q}_p - \boldsymbol{u}_p \right)$.
        - Update the seed number $s_p$.
        - Compute the average SG, $\bar{\tilde{\boldsymbol{g}}} = \frac{1}{P} \sum_p \tilde{\boldsymbol{g}}_p$, and broadcast it to the workers.
    **Workers** $p = 1, 2, \ldots, P$:
        - Receive average SG, $\bar{\tilde{\boldsymbol{g}}}$.
        - Update parameters according to the the preset training algorithm (SGD, ADAM, ...).

## 3.2 REDUCING COMMUNICATION OVERHEAD BY NESTED QUANTIZATION

It is well-known that correlated signals can be communicated more efficiently via distributed compression than the traditional entropy based coding Slepian & Wolf (1973). Nested Quantization has been proven to be a viable tool in communicating correlated data Zamir et al. (2002). Here, we propose to use nested quantization in distributed learning.

Let $(Q_1, Q_2)$ be a pair of nested quantizers with quantization step sizes $\Delta_1$ and $\Delta_2$, respectively and $0 < \alpha \le 1$ be a shrinkage factor whose value to be determined later. To quantize and transmit $x$, the worker first generates a random dither $u \sim \mathcal{U}[-\Delta_1/2, \Delta_1/2]$ and computes $t = \alpha x + u$. Then it quantizes and encodes it as

$$s = Q_1(t) - Q_2(t), \tag{6}$$

i.e., it transmits the position of the fine quantization bin relative to the coarse one (shown by indexes $-1, 0, 1$ in Fig. 3). At the receiver, by knowing $s$ alone, $x$ cannot be estimated reliably as multiple values can produce the same $s$. To resolve that ambiguity, it is required to know which coarse quantization bin $x$ belongs to. This is achieved by the help of the information provided by $y$, available at the receiver. $x$ is reconstructed from the received $s$ and using $y$ as follows:

$$r = s - u - \alpha y, \quad \hat{x} = y + \alpha(r - Q_2(r)). \tag{7}$$

*Note that quantizing $x$ does not require $y$, however estimating $x$ at the server depends on the information provided by $y$.*

Figure 3 shows an example of using nested quantization, where $\Delta_1 = 1$ and $\Delta_2 = 3$. Let $x = -4.2$ and $u = 0.3$ be the generated dither. Assume $\alpha = 1$, hence $s = Q_1(-3.9) - Q_2(-3.9) = -4 - (-3) = -1$ is the signal to be transmitted. Note that multiple points can produce the same $s$ with that dither signal, some are shown by ♦ in the figure, e.g., $-4.3., -1.3, 2.7, \ldots$ all leads to the same $s$. However, having access to $y = -3.4$ at the receiver can resolve the ambiguity. The value which resides in the *same coarse quantization bin* as $y$ is chosen, resulting in $\hat{x} = -4.3$. Note that in this nested quantization scheme, the output of quantizer is in

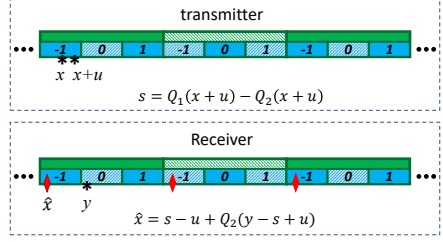

Figure 3: Nested quantization, $\Delta_1 = 1$, $\Delta_2 = 3$ and $\alpha = 1$.

$\{-1, 0, +1\}$. If we wanted to achieve the same accuracy with a single quantizer, we had to transmit

$s = -4$ instead of $s = -1$, increasing the number of bits depending on the range of $x$. For example, in Fig. 3, nested quantization reduces the range of quantization indexes from $\{-4, -3, \ldots, 4\}$ to $\{-1, 0, 1\}$, reduction by a factor of 3.

---

**Algorithm 2:** Distributed Training Using Nested Dithered Quantization of SG

---

**Workers** $p = 1, 2, \ldots, P$

    **if** $p \in \mathcal{P}_1$ **then**

        - Generate random dither $\boldsymbol{u}_p \sim \mathcal{U}[-\Delta_1/2, \Delta_1/2]$

        - Transmit $\boldsymbol{s}_p = Q_1(\boldsymbol{g}_p + \boldsymbol{u}_p)$

    **else if** $p \in \mathcal{P}_2$ **then**

        - Generate random dither $\boldsymbol{u}_p \sim \mathcal{U}[-\Delta_1^{(p)}/2, \Delta_1^{(p)}/2]$

        - Use nested dithered quantizer; transmit $\boldsymbol{s}_p = Q_{p_1}(\alpha_p \boldsymbol{g}_p + \boldsymbol{u}_p) - Q_{p_2}(\alpha_p \boldsymbol{g}_p + \boldsymbol{u}_p)$

**Server**

    - Compute $\overline{\overline{\boldsymbol{g}}} = \frac{1}{|\mathcal{P}_1|} \sum_{p \in \mathcal{P}_1} \tilde{\boldsymbol{g}}_p$ using received quantized gradients of workers in $\mathcal{P}_1$

    **for** $p \in \mathcal{P}_2$ **do**

        - Reproduce random dither $\boldsymbol{u}_p \sim \mathcal{U}[-\Delta_1^{(p)}/2, \Delta_1^{(p)}/2]$

        - Compute $\boldsymbol{r} = \boldsymbol{s}_p - \boldsymbol{u}_p - \alpha_p \overline{\overline{\boldsymbol{g}}}$

        - Decode the SG of worker $p$ as $\tilde{\boldsymbol{g}}_p = \overline{\overline{\boldsymbol{g}}} + \alpha_p(\boldsymbol{r} - Q_{p_2}(\boldsymbol{r}))$

        - Update $\overline{\overline{\boldsymbol{g}}}$ using $\tilde{\boldsymbol{g}}_p$.

---

Our proposed distributed training using nested dithered quantization is summarized in Alg. 2 for one iteration of training [4]. The stochastic gradient, computed by the $p$-th worker in a distributed training system, can be considered as a noisy estimate of the true gradient, i.e., $\boldsymbol{g}_p = \nabla_{\boldsymbol{w}} \mathcal{L} + \boldsymbol{\nu}_p$ where $\boldsymbol{\nu}_p$ is a zero-mean noise. However, as opposed to Zamir et al. (2002) and other similar works, the exact gradient $\nabla_{\boldsymbol{w}} \mathcal{L}$ is not available in distributed training. To overcome this issue, we propose to divide the workers into two groups. Set $\mathcal{P}_1$ of workers use DQSG with quantization step size $\Delta_1$, to provide an initial estimate for the true gradient. The parameter of the quantization and the number of workers in $\mathcal{P}_1$ are chosen such that the variance of averaged DQSG (see Lemma 3) becomes in an acceptable range, determined by Thm. 6. The workers in $\mathcal{P}_2$ use nested quantizer with step-sizes $(\Delta_1^{(p)}, \Delta_2^{(p)})$ and scale $\alpha_p$. To decode the received Nested Dithered Quantized SG (NDQSG), the receiver uses the average of all SGs already received and decoded from other workers, denoted by $\overline{\overline{\boldsymbol{g}}}$. We assume that the SG of the $p$-th worker can be modeled as $\boldsymbol{g}_p = \overline{\overline{\boldsymbol{g}}} + \boldsymbol{z}_p$, where $\boldsymbol{z}_p$ is an independent random noise. Hence, the nested quantization uses $\overline{\overline{\boldsymbol{g}}}$ at the receiver as the side information to compute $\widetilde{\boldsymbol{g}}_p$. To find the quantization parameters, we can use the following result;

**Theorem 6.** *If the SG at a worker is modeled by $\boldsymbol{g} = \overline{\overline{\boldsymbol{g}}} + \boldsymbol{z}$, $\mathbb{E}\left[z_i^2\right] = \sigma_z^2$, and the worker uses nested quantizer with parameters $\Delta_1$, $\Delta_2$ and $\alpha$, then with probability at least $1 - p$, $\widetilde{g}_i$ will be estimated correctly (i.e., $g_i$ and $\widetilde{g}_i$ are in the same coarse quantization bin), where*

$$p = \Pr\left(|\alpha z + u| > \frac{\Delta_2}{2}\right) \le \frac{\Delta_1^2}{3\Delta_2^2} + 4\alpha^2 \frac{\sigma_z^2}{\Delta_2^2}, \quad u \sim \mathcal{U}[-\Delta_1/2, \Delta_1/2]. \tag{8}$$

*Specially if $|\boldsymbol{z}| < \frac{\Delta_2 - \Delta_1}{2\alpha}$, then $p = 0$. In this case,*

$$\mathbb{E}\left[\|\widetilde{\boldsymbol{g}} - \boldsymbol{g}\|_2^2\right] = \alpha^2 \frac{\Delta_1^2}{12} + (1 - \alpha^2)^2 \sigma_z^2. \tag{9}$$

Note that setting $\alpha = 1$ or $\alpha = \sqrt{1 - \Delta_1^2/12\sigma_z^2}$ results in the same quantization variance as dithered quantization with step-size $\Delta_1$. However, nested quantization requires $\log_2(\Delta_{p_2}/\Delta_{p_1})$ bits to transmit each value, i.e., less than the ordinary quantization methods which requires almost $\log_2(2/\Delta_{p_1})$ bits.

---

[4]Note that we have ignored details on reproducing the pseudo-random sequences $\boldsymbol{u}_k$'s and updating seed numbers which are the same as in Alg. 1.

## 4 EXPERIMENTS

We examine the convergence and and the number of communication bits used by different learning algorithms based on DQSG and nested dithered quantized SG (NDQSG) for various number of workers, and compare them against the baseline (no quantization of gradients), one-bit quantization Seide et al. (2014), TernGrad Wen et al. (2017), and QSGD Alistarh et al. (2017). Although it is possible to evaluate the performance of the quantization and compression schemes in both synchronous and asynchronous settings, here we assume that the workers and server are synchronous. The main reason for such a setting is to cancel-out the performance degradation (in terms of training accuracy or speed) that may be caused by the stale gradients in asynchronous updates, and to solely investigate the effect of the quantization/compression algorithms.

We have considered three different models, a fully connected neural network with two hidden layers of sizes $300$ and $100$ over MNIST dataset (herein, referred to as FC-300-100), a Lenet-5 like convolutional network LeCun et al. (1998) over MNIST and a convolutional network Krizhevsky (2014) on Cifar10 (referred to as CifarNet), with SGD and Adam training algorithms. The initial learning rates for SGD and Adam are $0.01$ and $0.001$, respectively with decay rate $0.98$ per training epoch. The batch size is fixed at $256$ and divided evenly among the workers.

First, we observe that using entropy coding algorithms such as Adaptive Arithmetic Coding (ACC) can further reduce the communication bits for all schemes close to the entropy limit (within $5\%$ range). Therefore, it suffices to report both the number of raw communication bits from quantization as well as the resulting entropy of the bit-stream for comparison. Tables 1 and 2 show the raw (uncompressed) communication bits and the entropy per worker at each iteration of training, respectively. The communication bits of DQSGD and QSGD are close to each other. Although One-bit quantization requires less raw bits to transmit, it is less compressible, e.g., using entropy coding for Lenet, DQSGD would use 6 times less number of bits per iteration compared to one-bit quantization.

Table 1: Raw communication bits per worker (Kbits per iteration of training) for different networks

| Method | Baseline | DQSGD | QSGD | TernGrad | One-Bit |
|---|---|---|---|---|---|
| FC300-100 | 8531.5 | 422.8 | 422.8 | 426.2 | 342.6 |
| Lenet | 53227.8 | 2636.7 | 2636.7 | 2641.2 | 1897.8 |
| CifarNet | 34185.5 | 1690 | 1690 | 1692 | 1251 |

Table 2: Resulting bit stream per worker (Kbits per iteration of training) after entropy coding for different networks, 32 workers

| Method | DQSGD | QSGD | TernGrad | One-Bit |
|---|---|---|---|---|
| FC300-100 | 38.6 | 38.2 | 48.23 | 330 |
| Lenet | 299.7 | 307.3 | 438.2 | 1889 |
| CifarNet | 192.7 | 197 | 281 | 1241 |

Figure 4 shows the accuracy of the final trained model vs different number of workers for FC-300-100 and Lenet models. Table 3 shows the results for CifarNet model after 50 epochs ot training. From the simulations, it is seen that our proposed algorithm performs much better than the one-bit quantization method and is close to the baseline performance (non-quantized communication).

Moreover, in Fig. 5, we have compared the convergence rate of our dithered quantization scheme w.r.t. baseline (no quantization), one-bit quantization Seide et al. (2014) and QSGD Alistarh et al. (2017) for $4$ and $8$ workers. It is interesting to note that the dithered quantization improves the convergence of the training algorithm even when compared to the baseline (no quantization) in terms of number of training iterations. Although we do not have any analytic proof that using dithered quantization would *always* improve the convergence speed w.r.t. no quantization, because of the independency of the noise from the SGs in our proposed method, our method is likely to result in a better convergence property than the aforementioned techniques for complex training data Neelakantan et al. (2015);

Noh et al. (2017). On the other hand, as the number of workers increases, due to the averaging performed on the received quantized SGs, the noise would decrease proportionately and we expect the performance gap between different quantization methods eventually vanishes.

Table 3: Accuracy of CifarNet after 50 epochs of training, Adam training algorithm

| Method | Baseline | DQSG | QSG | TernGrad | One-Bit |
|---|---|---|---|---|---|
| 4 workers | 68.2 | 65.6 | 64.7 | 64.7 | 49.6 |
| 8 workers | 68.2 | 64.1 | 64.1 | 64 | 47.8 |

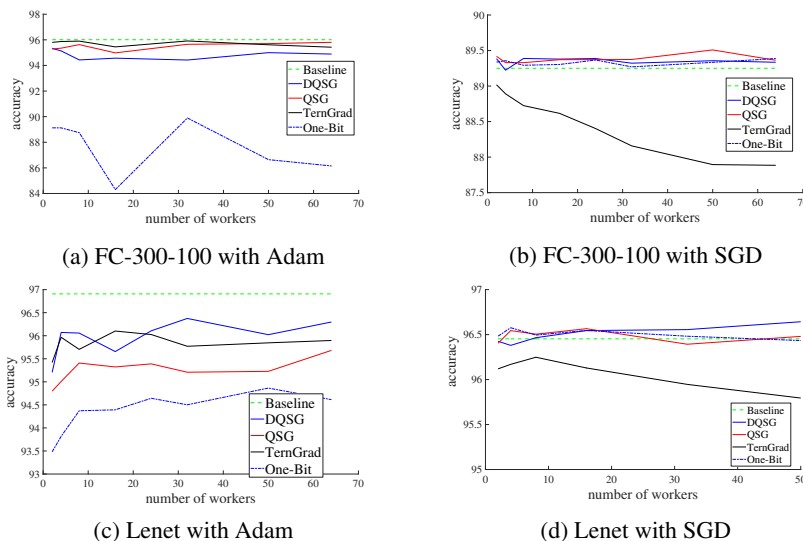

(a) FC-300-100 with Adam

(b) FC-300-100 with SGD

(c) Lenet with Adam

(d) Lenet with SGD

Figure 4: Accuracy of distributed training vs number of workers

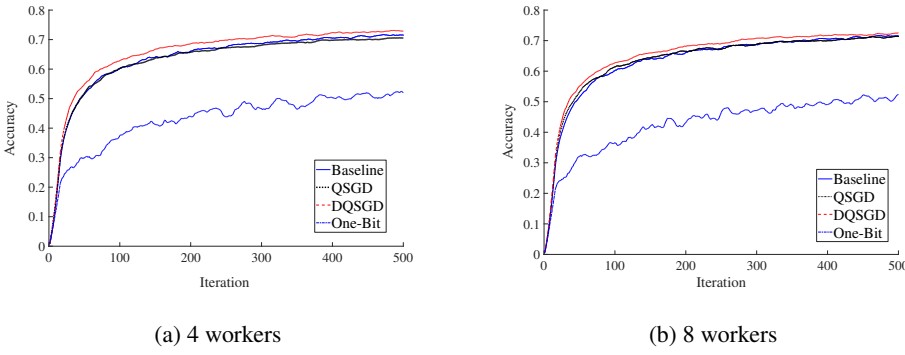

(a) 4 workers

(b) 8 workers

Figure 5: Comparison of convergence rate of distributed training of CifarNet with Adam algorithm

Next, we compare our nested dithered quantizer with the dithered quantization scheme. To have fair comparison, we chose the same expected accuracy for both quantization schemes. For DQSG, we chose $M = 2$, hence $\Delta = 0.5$ and the output of quantizer would be in $\{-2, \ldots, -2\}$. In NDQSG, for half of the workers, we divided the workers to two groups, half of the workers use DQSG with the same $\Delta$ and the other half, uses NDQSG with $\Delta_1 = 1/3$ and $\Delta_2 = 1$. Hence, the output of NDQSG quantizer is in $\{-1, 0, 1\}$. In Fig. 6 we compared the accuracy of NDQSG with DQSG and baseline training during training. As seen, the learning curve of NDQSG is almost the same as DQSG and the baseline. However, the communication bits are much less. For example, in training FC-300-100, with 2 level quantizers, QSG and DQSG requires 619.2 Kbits per worker to communicate, while NDQSG

reduces that to 422.8 Kbits, more than $30\%$ reduction in number of bits to communicate. The Same is true for the other considered neural networks.

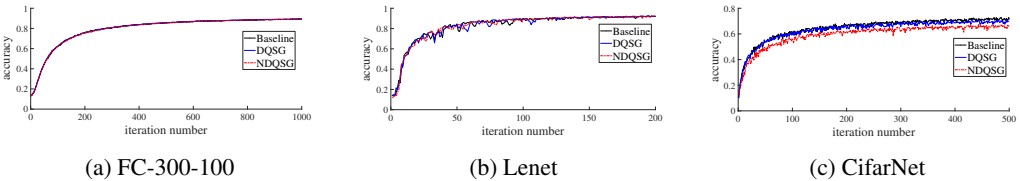

| (a) FC-300-100 | (b) Lenet | (c) CifarNet |

Figure 6: Accuracy of nested dithered quantization at each iteration of training for 8 workers

## 5 CONCLUSION

In this paper, first, we introduced DQSG, dithered quantized stochastic gradient, and showed that how it can reduce communication bits per training iteration both theoretically and via simulations, without affecting the accuracy of the trained model. Next, we explored the correlation that exists among the SGs computed by workers in a distributed system and proposed NDQSG, a nested quantization method for the SGs. Using theoretical analysis as well as simulations, we showed that NDSQG performs almost the same as DQSG in terms of accuracy and training speed, but with much fewer number of communication bits.

Finally, we would like to mention that although the simulations and analysis of the proposed distributed training method is done in synchronous training setup, it is applicable to the asynchronous training as well. Further, our nested quantization scheme can be easily extended to hierarchical distributed structures.

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

## A    PROOF OF LEMMA 2

Let $Q(\cdot)$ be a $2M+1$-level quantizer with step size $\Delta = 1/M$. Let $u \sim \mathcal{U}[-\Delta/2, \Delta/2]$ be the dither signal. Let $0 \le x \le 1$ be an arbitrary number. Assume that $l/M \le x < (l+1)/M$ and define $d = x - l/M$. Note that $0 \le d < \Delta$ and

$$P\left(Q(x+u) = \frac{l}{M}\right) = P\left(|x + u - l/M| \le \frac{\Delta}{2}\right) = P\left(u \le \frac{\Delta}{2} - d\right) = 1 - \frac{d}{\Delta} = 1 - Md.$$

Similarly, $P(Q(x+u) = (l+1)/M) = Md$. Comparing with stochastic quantizer, we see that they both assign the quantization points with the same probability. The case $x < 0$ can be verified similarly.

## B    PROOF OF LEMMA 3

To prove the unbiasedness, note that by Thm. 1, $e = Q(g/\kappa + u) - (g/\kappa + u)$ is independent from $g/\kappa$ and uniformly distributed over $[-\Delta/2, \Delta/2]$. On the other hand, $\tilde{g} = g + \kappa e$. Hence,

$$\mathbb{E}[\tilde{g}] = \mathbb{E}[g + \kappa e] \overset{(a)}{=} \mathbb{E}[g] + \mathbb{E}[\kappa]\,\mathbb{E}[e] \overset{(b)}{=} \nabla\mathcal{L},$$

where (a) is due to the fact that $\kappa = \|g\|_\infty$ is independent of $e$ and (b) because of unbiasedness of stochastic gradient and $e$ having mean zero.

For the variance,

$$\mathbb{E}\left[\|\tilde{g} - \nabla\mathcal{L}\|_2^2\right] = \mathbb{E}\left[\|g - \nabla\mathcal{L}\|_2^2\right] + \mathbb{E}\left[\|g\|_\infty^2\right]\text{Var}[e] \overset{(c)}{\le} \text{Var}[g] + \mathbb{E}\left[\|g\|_2^2\right]\frac{n\Delta^2}{12},$$

where (c) follows from $\mathbb{E}\left[\|e\|_2^2\right] = \sum_{i=1}^n \mathbb{E}\left[(e_i)^2\right] = n\Delta^2/12$, and $\|g\|_\infty \le \|g\|_2$.

To prove (3), note that for a given $g$,

$$\mathbb{E}\left[\|\tilde{g} - g\|_2^2 | g\right] = \|g\|_\infty^2 \frac{n\Delta^2}{12} \quad \Rightarrow \quad \mathbb{E}\left[\|\tilde{g} - g\|_2^2\right] = \mathbb{E}\left[\|g\|_\infty^2\right]\frac{n\Delta^2}{12}.$$

Let $\mu = \nabla_w\mathcal{L}$. The assumed model for $g$ implies that $g \sim \mathcal{N}(\mu, \sigma^2)$ and $\mathbb{E}\left[\|g - \nabla_w\mathcal{L}\|_2^2\right] = n\sigma^2$. For an arbitrary $t > 0$,

$$e^{t\,\mathbb{E}[\|g\|_\infty^2]} \overset{(d)}{\le} \mathbb{E}\left[e^{t\max_i |g_i|^2}\right] = \mathbb{E}\left[\max_i e^{t|g_i|^2}\right] \le \sum_i \mathbb{E}\left[e^{t|g_i|^2}\right],$$

where (d) follows from Jensen's inequality and definition of $\|\cdot\|_\infty$. Since $g_i \sim \mathcal{N}(\mu_i, \sigma^2)$,

$$\mathbb{E}\left[e^{t|g_i|^2}\right] = \frac{1}{\sqrt{1 - 2t\sigma^2}}\exp\left(\frac{\mu_i^2 t}{1 - 2t\sigma^2}\right), \quad \text{for } 0 \le t \le \frac{1}{2\sigma^2}.$$

Therefore,

$$e^{t\,\mathbb{E}[\|g\|_\infty^2]} \le \sum_{i=1}^n \mathbb{E}\left[e^{t|g_i|^2}\right] = \frac{1}{\sqrt{1 - 2t\sigma^2}}\sum_i \exp\left(\frac{t\mu_i^2}{1 - 2t\sigma^2}\right) \le \frac{n}{\sqrt{1 - 2t\sigma^2}}\exp\left(\frac{t\|\mu\|_\infty^2}{1 - 2t\sigma^2}\right)$$

$$\mathbb{E}\left[\|g\|_\infty^2\right] \le \frac{1}{t}\ln\left(\frac{n}{\sqrt{1 - 2t\sigma^2}}\right) + \frac{\|\mu\|_\infty^2}{1 - 2t\sigma^2}.$$

Setting $t = 1/4\sigma^2$ gives the desired bound in (3).

## C    A NOTE ON THM. 4

Because of the nature of quantization noise in our approach, the majority of convergence results with stochastic gradients can be readily applied to the DQSG. As an example, in this paper, we considered a result by Bottou (1998). To prove the convergence of (DQSGD), it suffices to show that there exists constants $A'$ and $B'$ such that $\mathbb{E}\left[\|\tilde{g}(w)\|_2^2\right] \le A' + B'\|w - w^*\|_2^2$;

$$\mathbb{E}\left[\|\tilde{g}(w)\|_2^2\right] \overset{(e)}{=} \mathbb{E}\left[\|\tilde{g} - g\|_2^2\right] + \mathbb{E}\left[\|g\|_2^2\right] = \frac{n\Delta^2}{12}\mathbb{E}\left[\|g\|_\infty^2\right] + \mathbb{E}\left[\|g\|_2^2\right] \le (1 + \frac{n\Delta^2}{12})\mathbb{E}\left[\|g\|_2^2\right].$$

Therefore, for $A' = (1 + \frac{n\Delta^2}{12})A$ and $B' = (1 + \frac{n\Delta^2}{12})B$, the DQSG is bounded and the theorem is proved following the same argument as in Bottou (1998).

# D   A NOTE ON THM. 5

This is a direct result of (Bubeck, 2015, §6). Note that $\mathbb{E}\left[\|\tilde{g} - \nabla_w \mathcal{L}\|_2^2\right] \leq V + \frac{n\Delta^2}{12}\mathbb{E}\left[\|g\|_2^2\right] \leq V(1 + n\Delta^2/12) + nB\Delta^2/12 = \sigma^2$ and since there are $P$ workers, the variance bound on $\widetilde{\overline{g}}$ would be $\sigma^2/P$. Then after $T$ iterations of (DQSGD) with step size $\eta_t = 1/(\ell + 1/\gamma)$ for $\gamma = \frac{R}{\sigma/\sqrt{P}}\sqrt{2/T}$,

$$\mathbb{E}\left[\mathcal{L}\left(\frac{1}{T}\sum_{t=1}^{T} w_t\right)\right] - \mathcal{L}(w^*) \leq R\sqrt{\frac{2\sigma^2}{PT}} + \frac{\ell R^2}{T}.$$

For $\epsilon < 0.2\sigma^2/PL$, set

$$T = 2.5\frac{R^2\sigma^2}{P\epsilon^2}.$$

Then, it can be easily verified that for the given step-size, the results hold.

# E   PROOF OF THM. 6

Let $e = \alpha g + u - Q_1(\alpha g + u)$ and $r = s - u - \alpha\widetilde{\overline{g}}$. Then,

$$\hat{g}_i = \widetilde{\overline{g}}_i + \alpha(r_i - Q_2(r_i)).$$

Since $\widetilde{\overline{g}}_i = g_i + z_i$, it can be shown that

$$r_i - Q_2(r_i) = \alpha z_i - e_i - Q_2(\alpha z_i - e_i).$$

Therefore,

$$\hat{g}_i = \widetilde{\overline{g}}_i + \alpha(\alpha z_i - e_i) - \alpha Q_2(\alpha z_i - e_i).$$

The correct decoding occurs when $Q_2(\alpha z_i - e_i) = 0$. Hence, the probability of correct recovery would be $1 - p$ where

$$p = \Pr\left(|\alpha_z + u| > \frac{\Delta_2}{2}\right), \quad u \sim \mathcal{U}[-\Delta_1/2, \Delta_1/2].$$

In that case,

$$\hat{g}_i = g_i - (\alpha e_i + (1 - \alpha^2)z_i).$$

Since $e_i \sim \mathcal{U}[-\Delta_1/2, \Delta_1/2]$ and $z_i$ are independent from each other and from $g_i$, simple calculations show that

$$\mathbb{E}\left[(\tilde{g}_i - g_i)^2\right] = \alpha^2\frac{\Delta_1^2}{12} + (1 - \alpha^2)^2\sigma_z^2.$$

