# OpenReview forum: "Nested Dithered Quantization for Communication Reduction in Distributed Training"
_ICLR.cc/2019/Conference_

### Official Review · AnonReviewer1 · 2018-10-30
**Establishes a useful connection between distributed optimization and dithered quantization**

**Rating:** 7
**Confidence:** 4

**Review:**

Authors establish a connection between communication reduction in distributed optimization and dithered quantization. This allows us to understand prior approaches in a new perspective, and also motivates authors to develop two new distributed training algorithms which communication overhead is significantly reduced. The first algorithm, DQSG, uses dithered quantization to reduce the communication bits. The second algorithm, NDQSG, uses nested dithered quantization to further reduce the amount of needed communication. The usefulness of these algorithms are empirically validated by computing the raw communication bits and average entropy of them. Therefore, dithered communication seems to provide both theory and algorithm which are useful.

The paper is clearly written. It provides a succinct review of dithered quantization and previous works, and figures provide a good insight into why the algorithm works, especially Figure 3.

Theorems in this paper are mostly about plugging in properties of dithered quantization into standard results in stochastic optimization, but they are still useful. The analysis of NDQSG does not seem to be as complete as that of DQSG, however. With NQSG, now workers are divided into two groups, and there would be an interesting tradeoff between assignments to these two: how should we balance two groups? This might be tricky to analyze, but it is still useful to clarify limitations and provide conjectures. At least, this could be analyzed empirically.

pros:
* establishing a connection to other topic of research often facilitates productive collaboration between two fields
* provides a new perspective to understand prior work
* provides new useful algorithms

cons:
* experiments were conducted on small models and small datasets
* unclear models are large enough to demonstrate the need for communication reduction; in other words, it is unclear wall-time would actually be reduced with these algorithms.

---

> ### Author Response · Authors · 2018-11-23
> **added simulations**
>
> Thanks for the comments and suggestions to improve the paper.
>
> We have added two more figures in the Experiments section to highlight how the dithered quantization scheme may improve the convergence rate of the distributed training. We would like to point out that for the presented simulation results, the convergence speed of our proposed method is better than the existing methods and including the baseline (communication without quantization). We argue that this is mainly due to the fact that our method basically adds a controlled independent noise to the stochastic gradients which may help with the convergence, consistent with the findings in [Neelakantan et al. 2015] and [Noh et al. 2017].

---

### Official Review · AnonReviewer3 · 2018-11-03
**Interesting paper but the contribution is not good enough**

**Rating:** 5
**Confidence:** 3

**Review:**

Overall, this paper is well written and clearly present their contribution.
Although the idea seems to be interesting and novel, but not enough evidence to prove the efficiency, from both theoretical and numerical perspective, even though many numerical experiments are proposed.
In general, this paper is high level in the articles assigned to me.

---

> ### Author Response · Authors · 2018-11-23
> **clarifications on the contribution of the work**
>
> We appreciate the reviewer's comment. However, as there is no direct comments or questions on the paper by the reviewer to justify his/her ratings, we just briefly state some of our contributions in this paper:
>
> 1- we have considered using dithered quantization for the communication of SG (or in general, parameters' updates) in a distributed training setting. The advantage of using our proposed qunatization scheme is that unlike all other existing methods where the added noise due to the quantization depends on the values of SG, here the quantization noise is inpendent of the SG values. This ensures that almost all existing convergence results on training with SG or its variants readily applicable to the quantized distributed training algorithm, without much modification. (see e.g., Thm. 4)
>
> 2- We have analyzed how the number of workers and quantization precision (or equivalently number of bits) affect the training times (see. Thm 5 and equation 5)
>
> 3- We provided a nested scheme to further reduce communication without sacrificing the precision of quantization. For example, theoretically we could achieve the accuracy of two bits quantization with only 1 bit in a distributed setting. (see Thm. 6 and the discussion after)
>
> 4- Finally, we provided some simulation results to experimentally verify the algorithm.
>
>
>
> Note that the proofs of all the claims and theoretical results are provided in the appendix.

---

### Official Review · AnonReviewer2 · 2018-11-05
**Nested Dithered Quantization for Communication Reduction in Distributed Training**

**Rating:** 5
**Confidence:** 4

**Review:**

In this paper, the authors propose to apply dithered quantization (DQ) to the stochastic gradients computed through the training process. Though an extra noise is added to the gradient, it improves the quantization error. Hence after the noise is removed at the update server, it achieves superior results when compared against unquantized baseline.

The authors also propose a nested scheme to further reduce communication cost.

This method strictly improves over previous approaches such as QSGD and TernGrad in terms of quantization error. However, the improved quantization performance does not show up in the experiments. In Table 3, it is clear that DQSG does not significantly improve over QSG and TernGrad once there are 8 workers. And they all use the same amount of bits in communication.

The proposed NDQSG though capable of reducing the communication cost by 30%, its accuracy on CIFAR-10 shows noticeable drop.

Overall, I think this method is promising, but further tuning is required to make it practical.

---

> ### Author Response · Authors · 2018-11-23
> **clarification on the simulations and comparisons w.r.t. other methods**
>
> We appreciate the reviewer's comments and suggestions.
>
> - Regarding the comparison of dithered quantization with One-bit, TernGrad and QSGD:
>
> We would like to point out that, due to indecency of noise from SGs in our scheme, the proposed distributed training algorithm is expected to behave consistently well irrespective of the database or the neural network.
>
> The results shown in the paper only reflects the final accuracy of the model after enough iterations of the training algorithm that the models have almost converged. Hence, they merely show the effect of the distributed training on the final accuracy, not how fast the models converge. To address this issue, we have added two figures in the paper showing the accuracy vs iteration for CIFAR10 with 4 and 8 workers, for the first 500 iterations of training. We would like to point out that (a) the convergence speed of our proposed method is better than the existing methods and including the baseline method (communication without quantization). We argue that this is mainly due to the fact that our method basically adds a controlled independent noise to the stochastic gradients which may help with the convergence, consistent with the findings in [Neelakantan et al. 2015] and [Noh et al. 2017]. (b) As the number of workers increases, since the average of received quantized SG is computed and used for training, the quantization noise would be decreased proportionately. Hence, the performance gap between almost all of the quantization methods for the distributed training will vanish eventually.
>
>
>
> - Regarding the comment on the NDQSG:
>
> Note that the main contribution of our work is the dithered quantization and its theoretical analysis in distributed training. However, we mentioned NDQSG to further reduce the communication by exploiting the correlations among SGs computed by the workers. The performance of NDQSG depends on the amount of correlation among SGs computed by the workers. The probability of error in distributed communication using NDQSG is bounded by Thm. 6 and equation 8. We advise on using NDQSG whenever the correlation is significant. As shown in Figues 6(a) and 6(b), when the correlations among the SGs computed by the workers are high, using NDQSG can reduce the communication cost. However, in Fig. 6(c), since the noise in SG computation is high, using NDQSG failes to estimate the true SG sometimes, adding some error into the estimation. This can slow down the convergence speed of the distributed training algorithm in some situations.

---

> > ### Comment · AnonReviewer2 · 2018-11-29
> > **Not enough for me to change my review.**
> >
> > Thanks for the rebuttal. I still find that the practical impact of this method is not clear.
> >
> > For one thing, this method needs low level change in the training framework. And if there is no clear quality or performance gain, it becomes hard to justify the extra complexity.

---

> > > ### Author Response · Authors · 2018-11-30
> > > **on the complexity of the algorithm**
> > >
> > > Thanks for the feedback.
> > >
> > > I would like to mention that the complexity of the dithered quantization at the workers is similar to the other stochastic quantization methods such as TernGrad and QSGD. Hence, at the worker side, the complexity of the algorithm would be the same.
> > > However, for dequantization, our method requires the server (or aggregation node) to regenerate the random numbers and then taking their average, which is not of much computational complexity. This can be done while the workers are computing their SGs and the server is waiting to receive their data. Hence, we believe that at each iteration of distributed training, using dithered quantization would not increase the complexity or the required time for computing the average of received SGs from workers.

---

### Meta-Review · Area_Chair1 · 2018-12-13

**Confidence:** 5
**Recommendation:** Reject

**Metareview:**

The reviewers found that the paper needs more compelling empirical study.